# FSEO: Few-Shot Evolutionary Optimization via Meta Learning for Expensive Multi-Objective Optimization

**Xunzhao Yu**

Department of Economics, University of Warwick

`Xunzhao.Yu@warwick.ac.uk`

## Abstract

Meta-learning has been demonstrated to be useful to improve the sampling efficiency of Bayesian optimization (BO) and surrogate-assisted evolutionary algorithms (SAEAs) when solving expensive optimization problems (EOPs). Existing studies mainly focus on either combinations of existing meta-learning modeling methods with optimization algorithms, or the development of meta-learning acquisition functions for specific meta BO. However, the meta-learning models used in the literature are not designed for optimization purposes, and the generalization ability of meta-learning acquisition functions is limited. In this work, we develop a novel architecture of meta-learning model for optimization purposes and propose a generalized few-shot evolutionary optimization (FSEO) framework to solve EOPs. We focus on the scenario of expensive multi-objective EOPs (EMOPs) in the context of few-shot optimization as there are few studies on it and its high requirement on surrogate modeling performance. The surrogates in FSEO framework combines neural network with Gaussian Processes (GPs), their network parameters and some parameters of GPs represent task-independent experience and are meta-learned across related optimization tasks, the remaining GPs parameters are task-specific parameters that represent unique features of the target task. We demonstrate that FSEO is able to improve the sampling efficiency of existing SAEAs on EMOPs.

## 1 Introduction

Expensive optimization problems (EOPs) aim to find as good as possible solutions within a budget of limited solution evaluations. Conventional Bayesian optimization (BO) and surrogate-assisted evolutionary algorithms (SAEAs) have been widely used to solve EOPs, but they train surrogate models from scratch. To further improve sampling efficiency and optimization performance, many efforts have been made to pre-train surrogates with the prior experience gained from related optimization tasks, resulting in experience-based optimization algorithms [2, 24, 38, 37].

**Scope.** This work considers solving EOPs in the context of few-shot problems [7, 44], where plenty of expensive related tasks are available and each of them can provide a small dataset for experience learning. Therefore, many experience-based optimization approaches, such as multi-tasking optimization [47, 3, 52] and transfer optimization [37, 21, 20], are **not** considered as they cannot learn experience from small related tasks (a detailed clarification of differences between these concepts is available in Appendix A.1). In comparison, meta-learning [17] has been proven to be powerful in solving few-shot problems, leading to a new subcategory of experience-based optimization, namely few-shot optimization (FSO) [50].

**Motivation.** Existing studies on FSO are mainly few-shot Bayesian optimization (FSBO) where meta-learning approaches are combined with BO to solve EOPs with only one objective. These studies either employ meta-learning models from the literature directly or focus on the meta-learning of acquisition functions (AFs) that are customized for BO. In this paper, we develop a novel meta-

39th Conference on Neural Information Processing Systems (NeurIPS 2025).

learning architecture for optimization purposes to enhance modeling performance and propose a generalized few-shot evolutionary optimization (FSEO) framework to address EOPs from the perspective of SAEAs. We demonstrate the generality and applicability of FSEO on multi-objective EOPs (EMOPs). FSO on EMOPs has been limited studied but EMOPs have a higher requirement on modeling performance than expensive single-objective optimization. Major contributions are summarized as follows.

- A novel meta-learning method, namely Meta Deep Kernel Learning (MDKL), is developed to gain prior experience from related expensive tasks. Our model architecture and parameter designs make it possible to generate a regression-based surrogate on the prior experience and then continually adapt the surrogate to approximate the target task.

- We propose a FSEO framework to solve EOPs from the perspective of SAEAs. FSEO framework is applicable to regression-based SAEAs since FSEO embed our meta-learning models in these SAEAs as their surrogates. In addition, an update strategy is designed to constantly adapt surrogates during optimization. Note that our FSEO framework is a general framework but we focus on its performance on EMOPs in this paper.

- Experiments are conducted on EMOPs to show that our FSEO framework is effective. Our comprehensive ablation studies reveal the influence of several factors on FSEO performance and provide empirical guidance on the application of our FSEO framework.

## 2 Related Work

Experience-based optimization can be divided into several subcategories according to the techniques of learning prior experience from related tasks. A detailed classification and discussion on these subcategories is available in Appendix A.1. This subsection focuses on related work on FSO.

Wistuba [50] firstly employed meta-learning for few-shot optimization on hyperparameter optimization (HPO) problems. Subsequent FSO related studies can be grouped into two categories: The first category focuses on improving the performance of few-shot optimization, either by employing different models [27] or by developing novel acquisition functions for BO [18]. The second category extends few-shot optimization to more complex optimization problems [56] or applies it to new domains [6, 8]. In addition, FSO studies in the literature can also be categorized according to their model architectures. Most studies meta-learn parameters for Gaussian Processes (GPs) [48], namely FSBO or Meta Bayesian Optimization (MBO) [34, 45, 29, 40]. In addition, [27] meta-learns with transformer neural processes and [50, 8] meta-learn parameters for the architecture of deep kernel learning (DKL) [49]. The MDKL model in our FSEO belongs to the last category as its model architecture is relevant to DKL.

Our work differs from existing studies in three points: Firstly, the novel architecture of meta-learning model for optimization purposes. Many studies [50] use existing meta-learning models [30] as their surrogates. During the optimization process, these surrogates make predictions with newly observed data, which is a kind of data adaptation rather than a model parameter adaptation. The parameters in these models are trained and fixed before the optimization process begins, no further parameter adaptations are made during the optimization, as these meta-learning models are originally designed for regression or classification tasks rather than optimization tasks. In comparison, we develop a meta-learning model, MDKL, for optimization purposes. MDKL has a novel model architecture with explicit task-specific parameters, which allows continual adaptations of model parameters and thus improves modeling performance during optimization. Secondly, the generality and broad applicability of FSEO. Existing works are mainly customized for specific algorithms or optimization problems. For example, the meta-learning settings for AFs [46] are not applicable to the SAEAs without AFs. However, our FSEO works on the meta-learning of surrogates and is applicable to various SAEAs, so our work broadens the scope of existing FSO research. A detailed discussion between BO and SAEA is presented in Appendix A.2. In addition, most existing FSO studies investigated only global optimization, leaving other optimization scenarios such as EMOP still awaiting investigation. In contrast, as our MDKL is designed for optimization and is capable of continually adaptation, we focus on EMOPs which require more effective models than global optimization. Lastly, in-depth ablation studies are lacking in the literature, making it unclear which factors affect the performance of FSO. Our extensive ablation studies fill this gap and we conclude some empirical rules to improve the performance of FSO.

## 3 Background

Preliminaries about meta-learning and DKL are given here. The former is the method of experience learning, the latter is the underlying structure of experience representation.

**Meta-Learning in Few-Shot Problems.** In the context of few-shot problems, we have plenty of related tasks, each task $\mathcal{T}$ contributes a couple of small datasets $\mathcal{D} = \{(\mathcal{S}, \mathcal{Q})\}$, namely support dataset $\mathcal{S}$ and query dataset $\mathcal{Q}$, respectively. After learning from datasets of random related tasks, a support set $\mathcal{S}_*$ from a new unseen task $\mathcal{T}_*$ is given and one is asked to estimate the labels or values of a query set $\mathcal{Q}_*$. The problem is called 1-shot or 5-shot when only 1 data point or 5 data points are provided in $\mathcal{S}_*$. A comprehensive definition of few-shot problems is available in [7, 44].

Meta-learning methods have been widely used to solve few-shot problems [44]. They learn domain-specific features that are shared among related tasks as experience, such experience is used to understand and interpret the data collected from new tasks encountered in the future.

**Deep Kernel Learning (DKL).** DKL aims to construct kernels that encapsulate the expressive power of deep architectures for GPs. To create expressive and scalable closed-form covariance kernels, DKL combines the non-parametric flexibility of kernel methods and the structural properties of deep neural networks. In practice, a deep kernel $k(\mathbf{x}^i, \mathbf{x}^j | \boldsymbol{\gamma})$ transforms the inputs $\mathbf{x}$ of a base kernel $k(\mathbf{x}^i, \mathbf{x}^j | \boldsymbol{\theta})$ through a non-linear mapping given by a deep architecture $\phi(\mathbf{x} | \mathbf{w}, \mathbf{b})$:

$$k(\mathbf{x}^i, \mathbf{x}^j | \boldsymbol{\gamma}) = k(\phi(\mathbf{x}^i | \mathbf{w}, \mathbf{b}), \phi(\mathbf{x}^j | \mathbf{w}, \mathbf{b}) | \boldsymbol{\theta}), \tag{1}$$

where $\boldsymbol{\theta}$ and $(\mathbf{w}, \mathbf{b})$ are parameter vectors of the base kernel and the deep architecture, respectively. $\boldsymbol{\gamma} = \{\boldsymbol{\theta}, \mathbf{w}, \mathbf{b}\}$ is the set of all the parameters in this deep kernel. Note that in DKL, all parameters $\boldsymbol{\gamma}$ of a deep kernel $k(\mathbf{x}^i, \mathbf{x}^j | \boldsymbol{\gamma})$ are learned jointly by using the log marginal likelihood function of GPs as the loss function. Such a joint learning strategy has been shown to make a DKL algorithm outperform a combination of a deep neural network and a GP model, where a trained GP model is applied to the output layer of a trained deep neural network [49].

**Meta-Learning on DKL.** An important distinction between DKL algorithms and the applications of meta-learning to DKL is that DKL algorithms learn their deep kernels from single tasks instead of collections of related tasks. This difference alleviates two drawbacks of single task DKL [41]: First, the scalability of deep kernels is no longer an issue as datasets in meta-learning are small. Second, the risk of overfitting decreases since diverse data points are sampled across tasks.

## 4 Few-Shot Evolutionary Optimization (FSEO) Framework

In this paper, $\mathcal{T}_*$ denotes the target optimization task, and plenty of small datasets $\mathcal{D}_i$ sampled from related tasks $\mathcal{T}_i$ are available for experience learning. A complete list of notations is available at the beginning of the Appendix.

### 4.1 Overall Working Mechanism

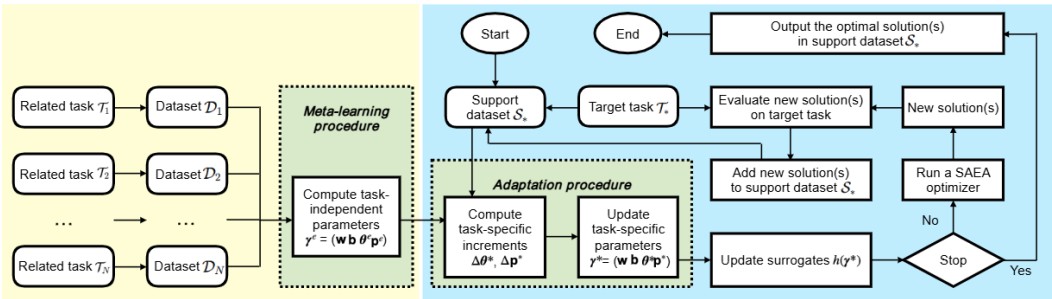

Figure 1: Diagram of our FSEO framework. Methods for handling multiple objectives or constraints are dependent on the module 'SAEA optimizer'.

As illustrated in Fig. 1, all modules covering the optimization of target task $\mathcal{T}_*$ are included in the blue block. The modules included in the yellow block are associated with related tasks $\mathcal{T}_i$ and

**Algorithm 1** FSEO Framework.

---

1: **Input:** $\mathcal{D}_i$: Datasets collected from related tasks $\mathcal{T}_i$, i=$\{1,\ldots,N\}$; $N_m$: Number of subsets $\mathcal{D}_m$ for meta-learning; $|\mathcal{D}_m|$: Size of subsets $\mathcal{D}_m$. $|\mathcal{D}_m| \leq |\mathcal{D}_i|$ due to $\mathcal{D}_m \subseteq \mathcal{D}_i$; Batch size $B$; Surrogate learning rates $\alpha, \beta$; Target task $\mathcal{T}_*$; An SAEA optimizer $Opt$; Fitness evaluation budget $FE_{max}$.
2: Experience $\boldsymbol{\gamma}^e \leftarrow$ Meta-learning($\mathcal{D}_i, N_m, |\mathcal{D}_m|, B, \alpha$). /*Alg. 2.*/
3: $\mathcal{S}_* \leftarrow$ Sampling $1d$ solutions from $\mathcal{T}_*$.
4: $h(\boldsymbol{\gamma}^*) \leftarrow$ Adaptation($\boldsymbol{\gamma}^e, \mathcal{S}_*, \beta$). /*Initialize surrogate.*/
5: Set evaluation counter $FE = |\mathcal{S}_*|$.
6: **while** $FE < FE_{max}$ **do**
7:    Candidate solution(s) $\mathbf{x}^* \leftarrow$ Surrogate-assisted optimization ($Opt, h(\boldsymbol{\gamma}^*)$).
8:    $f(\mathbf{x}^*) \leftarrow$ Evaluate $\mathbf{x}^*$ on $\mathcal{T}_*$.
9:    $\mathcal{S}_* \leftarrow \mathcal{S}_* \cup \{(\mathbf{x}^*, f(\mathbf{x}^*))\}$.
10:    $h(\boldsymbol{\gamma}^*) \leftarrow$ Update($\boldsymbol{\gamma}^*, \mathcal{S}_*, \beta$). /*Alg. 4.*/
11:    Update $FE$.
12: **end while**
13: **Output:** Optimal solutions in $\mathcal{S}_*$.

---

experience learning, which distinguishes our FSEO framework from conventional SAEAs and BO. The MDKL surrogate modeling method consists of two procedures: the meta-learning procedure and the adaptation procedure. The former learns prior experience from $\mathcal{T}_i$, the latter uses experience to adapt surrogates to fit $\mathcal{T}_*$. The framework of FSEO is depicted in Alg. 1, it consists of the following major steps.

1. **Experience learning**: Before expensive optimization begins, a meta-learning procedure is performed to train task-independent parameters $\boldsymbol{\gamma}^e$ for MDKL surrogates (line 2). $N_m$ datasets $\{\mathcal{D}_{m1}, \ldots, \mathcal{D}_{mN_m}\}$ collected from $N$ related tasks $\{\mathcal{T}_1, \ldots, \mathcal{T}_N\}$ are used to train $\boldsymbol{\gamma}^e$. $\boldsymbol{\gamma}^e$ is the experience that represents the domain-specific features of related tasks.

2. **Initialize surrogates with experience**: Optimization begins when a target optimization task $\mathcal{T}_*$ is given. An initial dataset $\mathcal{S}_*$ is sampled (line 3) to adapt task-specific parameters $\boldsymbol{\gamma}^*$ on the basis of experience $\boldsymbol{\gamma}^e$. After that, MDKL surrogates are updated (line 4).

3. **Reproduction**: MDKL surrogates $h(\boldsymbol{\gamma}^*)$ are combined with an SAEA optimizer $Opt$ to search for optimal solution(s) $\mathbf{x}^*$ on $h(\boldsymbol{\gamma}^*)$ (line 7). This is implemented by replacing the original (regression-based) surrogates in an SAEA with $h(\boldsymbol{\gamma}^*)$.

4. **Update archive and surrogates**: New optimal solution(s) $\mathbf{x}^*$ is evaluated on target task $\mathcal{T}_*$ (line 8). The evaluated solutions will be added to dataset $\mathcal{S}_*$ (line 9) which serves as an archive. Then, surrogate adaptation is triggered, surrogates $h(\boldsymbol{\gamma}^*)$ are updated (line 10).

5. **Stop criterion**: Once the evaluation budget has run out, the evolutionary optimization process terminates, outputting the optimal solutions in dataset $\mathcal{S}_*$. Otherwise, the algorithm returns to Step 3.

### 4.2 Learning and Using Experience by MDKL

In MDKL, the domain-specific features of related tasks are used as experience, which are represented by the task-independent parameters $\boldsymbol{\gamma}^e$ learned across related tasks. To make MDKL more capable of expressing complex domain-specific features, the base kernel $k(\mathbf{x}^i, \mathbf{x}^j | \boldsymbol{\theta})$ in GP is combined with a neural network $\phi(\mathbf{w}, \mathbf{b})$ to construct a deep kernel (see Eq.(1)). The modeling of an MDKL model consists of two procedures: the meta-learning procedure and the adaptation procedure. To illustrate them clearly, we present frameworks for both procedures and explain them in detail.

**Meta-learning procedure: Learning experience**
Our MDKL model uses the kernel in [22] as its base kernel:

$$k(\mathbf{x}^i, \mathbf{x}^j | \boldsymbol{\theta}, \mathbf{p}) = \exp(-\sum_{k=1}^{d} \theta_k |x_k^i - x_k^j|^{p_k}).$$
(2)

**Algorithm 2** Meta-learning($\mathcal{D}_i, N_m, |\mathcal{D}_m|, B, \alpha$)

---

1: **Input:** $\mathcal{D}_i$: Datasets collected from related tasks $\mathcal{T}_i$, i={1, ..., N}; $N_m$: Number of subsets $\mathcal{D}_m$ for meta-learning; $|\mathcal{D}_m|$: Size of subsets $\mathcal{D}_m$. $|\mathcal{D}_m| \leq |\mathcal{D}_i|$ due to $\mathcal{D}_m \subseteq \mathcal{D}_i$; Batch size $B$; Learning rate for priors $\alpha$.
2: Randomly initialize $\mathbf{w}, \mathbf{b}, \boldsymbol{\theta}^e, \mathbf{p}^e$.
3: Set the number of update iterations U = $N_m/B$.
4: **for** $j = 1$ to $U$ **do**
5:     $\{D'_1, ..., D'_B\} \leftarrow$ Randomly select a batch of datasets from $\{\mathcal{D}_1, ..., \mathcal{D}_N\}$.
6:     **for all** $D'_i$ in the batch **do**
7:         $\mathcal{D}_{mi} \leftarrow$ A subset of size $|\mathcal{D}_m|$ from $D'_i$.
8:         Initialize task-specific increment $\Delta\boldsymbol{\theta}^i, \Delta\mathbf{p}^i$.
9:         Compute task-specific parameters: $\boldsymbol{\theta}^i = \boldsymbol{\theta}^e + \Delta\boldsymbol{\theta}^i, \mathbf{p}^i = \mathbf{p}^e + \Delta\mathbf{p}^i$.
10:        Obtain deep kernel $k(\mathbf{x}^i, \mathbf{x}^j | \boldsymbol{\gamma})$ based GP: $h(\boldsymbol{\gamma})$, where $\boldsymbol{\gamma} = \{\mathbf{w}, \mathbf{b}, \boldsymbol{\theta}^i, \mathbf{p}^i\}$ (Eq.(3)).
11:        Compute the loss function $\mathcal{L}(\mathcal{D}_{mi}, h(\boldsymbol{\gamma}))$ (Eq.(4)).
12:     **end for**
13:     Update $\mathbf{w}, \mathbf{b}, \boldsymbol{\theta}^e, \mathbf{p}^e$ via gradient descent: $\alpha \bigtriangledown \mathcal{L}(\mathcal{D}_{mi}, h(\boldsymbol{\gamma}))$ (Eq.(6)).
14: **end for**
15: **Output:** Task-independent parameters: $\boldsymbol{\gamma}^e = \{\mathbf{w}, \mathbf{b}, \boldsymbol{\theta}^e, \mathbf{p}^e\}$.

---

Therefore, the deep kernel will be:

$$k(\mathbf{x}^i, \mathbf{x}^j | \boldsymbol{\gamma}) = \exp(-\sum_{k=1}^{d} \theta_k |\phi(x_k^i | \mathbf{w}, \mathbf{b}) - \phi(x_k^j | \mathbf{w}, \mathbf{b})|^{p_k}), \tag{3}$$

where $\boldsymbol{\gamma} = \{\mathbf{w}, \mathbf{b}, \boldsymbol{\theta}, \mathbf{p}\}$ is a set of deep kernel parameters. $\phi, \mathbf{w}$ and $\mathbf{b}$ denote the neural network and its parameters (see Eq.(1)). $\boldsymbol{\theta}, \mathbf{p}$ are parameters of base kernel, details on alternative base kernels are available in [48].

The aim of the meta-learning procedure is to learn experience $\boldsymbol{\gamma}^e$ from related tasks $\{\mathcal{T}_1, ..., \mathcal{T}_N\}$, including neural network parameters $\mathbf{w}, \mathbf{b}$, and task-independent base kernel parameters $\boldsymbol{\theta}^e, \mathbf{p}^e$. The pseudo-code of the meta-learning procedure is presented in Alg. 2. Ideally, experience $\boldsymbol{\gamma}^e$ is learned from plenty of ($N_m$) small datasets $\mathcal{D}_m$ collected from different related tasks. However, in practice, the number of available related tasks $N$ may be much smaller than $N_m$. Hence, meta-learning is conducted gradually over $U$ update iterations (line 3). During each update iteration, a small batch of related tasks contribute $B$ small datasets $\{\mathcal{D}_{m1}, ..., \mathcal{D}_{mB}\}$ for meta-learning purposes (lines 5 and 7). Note that if $N < N_m$, a related task $\mathcal{T}_i$ can be used multiple times in the meta-learning procedure.

For a given dataset $\mathcal{D}_{mi}$, we denote $\boldsymbol{\theta}^i = \boldsymbol{\theta}^e + \Delta\boldsymbol{\theta}^i$ and $\mathbf{p}^i = \mathbf{p}^e + \Delta\mathbf{p}^i$ as the task-specific kernel parameters, where $\Delta\boldsymbol{\theta}^i, \Delta\mathbf{p}^i$ are the distance we need to move from the task-independent parameters to the task-specific parameters (line 9). The loss function $\mathcal{L}$ of MDKL is the negative log-likelihood function, where the likelihood is defined as follows [22]:

$$\frac{1}{(2\pi)^{n/2}(\sigma^2)^{n/2}|\mathbf{R}|^{1/2}} exp[-\frac{(\mathbf{y} - \mathbf{1}\mu)^T \mathbf{R}^{-1}(\mathbf{y} - \mathbf{1}\mu)}{2\sigma^2}], \tag{4}$$

where $|\mathbf{R}|$ is the determinant of the correlation matrix $\mathbf{R}$, each element in the matrix is computed using Eq.(3). $\mathbf{y}$ is the fitness vector of $\mathcal{D}_{mi}$. Mean $\mu$ and variance $\sigma^2$ of the prior distribution can be estimated by:

$$\hat{\mu} = \frac{\mathbf{1}^T \mathbf{R}^{-1} \mathbf{y}}{\mathbf{1}^T \mathbf{R}^{-1} \mathbf{1}}, \qquad \hat{\sigma} = \frac{1}{n}(\mathbf{y} - \mathbf{1}\hat{\mu})^T \mathbf{R}^{-1}(\mathbf{y} - \mathbf{1}\hat{\mu}). \tag{5}$$

Experience $\boldsymbol{\gamma}^e = \{\mathbf{w}, \mathbf{b}, \boldsymbol{\theta}^e, \mathbf{p}^e\}$ is updated by gradient descent (line 13), take $\boldsymbol{\theta}^e$ as an example:

$$\boldsymbol{\theta}^e \leftarrow \boldsymbol{\theta}^e - \frac{\alpha}{B}\sum_{i=1}^{B} \bigtriangledown_{\boldsymbol{\theta}^e} \mathcal{L}(\mathcal{D}_{mi}, h(\boldsymbol{\gamma})). \tag{6}$$

After $U$ iterations, $\boldsymbol{\gamma}^e$ has been sufficiently trained by $N_m$ small datasets $\mathcal{D}_m$ and will later be used in target task $\mathcal{T}_*$.

**Adaptation procedure: Using experience**
The meta-learning of experience $\boldsymbol{\gamma}^e$ enables MDKL to handle a family of related tasks in general. To

effectively approximate a specific task $\mathcal{T}_*$, surrogate $h(\boldsymbol{\gamma}^e)$ needs to adapt task-specific increments $\Delta\boldsymbol{\theta}^*$ and $\Delta\mathbf{p}^*$ in the way described in Alg. 3. A diagram of the deep kernel implemented in our MDKL model is illustrated in Fig. 2: From Fig. 2, it is clear that task-independent parameters $\boldsymbol{\gamma}^e = \{\mathbf{w}, \mathbf{b}, \boldsymbol{\theta}^e, \mathbf{p}^e\}$ are trained on meta data $\mathcal{D}_i$. During the optimization process, MDKL adapts task-specific increments $\Delta\boldsymbol{\theta}^*, \Delta\mathbf{p}^*$ (Algorithm 8, line 3) and combines them with experience $\boldsymbol{\theta}^e$, resulting in task-specific parameters $\boldsymbol{\theta}^*, \mathbf{p}^*$. Hence, the deep kernel parameter $\boldsymbol{\gamma}^* = \{\mathbf{w}, \mathbf{b}, \boldsymbol{\theta}^*, \mathbf{p}^*\}$ is available. By invoking Eq. 5, the prior distribution of MDKL is estimated for the following surrogate prediction procedure.

---

**Algorithm 3** Adaptation($\boldsymbol{\gamma}^*, \mathcal{S}_*, \beta$)

1: **Input:** Current surrogate parameters $\boldsymbol{\gamma}^*$; A dataset $\mathcal{S}_*$ sampled from target task $\mathcal{T}_*$ (Archive); Learning rate for adaptation $\beta$.
2: **if** $\boldsymbol{\gamma}^* == \boldsymbol{\gamma}^e$ **then**
3:     Initialize task-specific increments $\Delta\boldsymbol{\theta}^*, \Delta\mathbf{p}^*$.
4:     Compute task-specific parameters: $\boldsymbol{\theta}^* = \boldsymbol{\theta}^e + \Delta\boldsymbol{\theta}^*$, $\mathbf{p}^* = \mathbf{p}^e + \Delta\mathbf{p}^*$.
5:     Obtain deep kernel $k(\mathbf{x}^i, \mathbf{x}^j | \boldsymbol{\gamma}^*)$ based GP: $h(\boldsymbol{\gamma}^*)$, where $\boldsymbol{\gamma}^* = \{\mathbf{w}, \mathbf{b}, \boldsymbol{\theta}^*, \mathbf{p}^*\}$ (Eq.(3)).
6: **end if**
7: Compute the loss function $\mathcal{L}(\mathcal{S}_*, h(\boldsymbol{\gamma}^*))$ (Eq.(4)).
8: Update $\Delta\boldsymbol{\theta}^*, \Delta\mathbf{p}^*$ using gradient descent: $\beta\bigtriangledown \mathcal{L}(\mathcal{S}_*, h(\boldsymbol{\gamma}^*))$.
9: **Output:** Adapted MDKL $h(\boldsymbol{\gamma}^*)$.

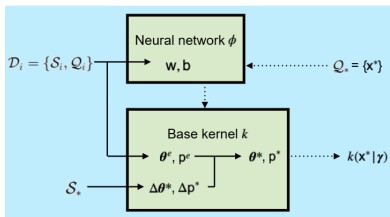

Figure 2: Diagram of our deep kernel implementation. The solid lines depict the training process, the dotted lines depict the inference process. $\mathcal{Q}_*$ denotes query samples to be evaluated on our surrogates.

---

**Surrogate prediction.** Due to the nature of a GP, when predicting the fitness of a solution $\mathbf{x}^*$, an MDKL surrogate produces a predictive Gaussian distribution $\mathcal{N}(\hat{y}(\mathbf{x}^*), \hat{s}^2(\mathbf{x}^*))$, the predicted mean $\hat{y}(\mathbf{x}^*)$ and covariance $\hat{s}^2(\mathbf{x}^*)$ are specified as [22]:

$$\hat{y}(\mathbf{x}^*) = \mu + \mathbf{r}'\mathbf{R}^{-1}(\mathbf{y} - \mathbf{1}\mu), \qquad \hat{s}^2(\mathbf{x}^*) = \sigma^2(1 - \mathbf{r}'\mathbf{R}^{-1}\mathbf{r}), \qquad (7)$$

where $\mathbf{r}$ is a correlation vector consisting of covariances between $\mathbf{x}^*$ and $\mathcal{S}_*$, other variables are explained in Eq.(4).

## 4.3 Surrogate Update Strategy

This subsection describes the update strategy in our FSEO framework. To properly integrate experience and data from $\mathcal{T}_*$, our update strategy is designed to determine whether an MDKL surrogate should be adapted in the current iteration or not, ensuring an optimal surrogate update frequency.

As illustrated in Alg. 4, the surrogate update begins when a new optimal solution(s) has been evaluated on expensive functions and an updated archive $\mathcal{S}_*$ is available. For a given surrogate $h(\boldsymbol{\gamma}^*)$, its mean squared error (MSE) on $\mathcal{S}_*$ is selected as the update criterion: If the MSE after an adaptation $e_1$ (line 4) is larger than the MSE without an adaptation $e_0$ (line 2), then the surrogate will roll back to the status before the adaptation. This indicates the surrogate update has been refused and $h(\boldsymbol{\gamma}^*)$ will not be adapted in the current iteration. Otherwise, the adapted surrogate will be chosen (line 6). Note that no matter whether surrogate adaptations are accepted or refused, the resulting surrogates will be treated as updated surrogates, which are employed to assist the SAEA optimizer in the next iteration.

---

**Algorithm 4** Update($\boldsymbol{\gamma}^*, \mathcal{S}_*, \beta$)

1: **Input:**
    Current surrogate parameters $\boldsymbol{\gamma}^*$;
    Updated archive $\mathcal{S}_*$;
    Learning rate for further adaptations $\beta$.
2: $e_0 \leftarrow \text{MSE}(h(\boldsymbol{\gamma}^*), \mathcal{S}_*)$.
3: $h(\boldsymbol{\gamma}') \leftarrow \text{Adaptation}(\boldsymbol{\gamma}^*, \mathcal{S}_*, \beta)$.
    /*Temporary surrogate, Alg. 3.*/
4: $e_1 \leftarrow \text{MSE}(h(\boldsymbol{\gamma}'), \mathcal{S}_*)$.
5: **if** $e_0 > e_1$ **then**
6:     update $\boldsymbol{\gamma}^* = \boldsymbol{\gamma}'$, obtain new $h(\boldsymbol{\gamma}^*)$.
7: **end if**
8: **Output:** Surrogate $h(\boldsymbol{\gamma}^*)$.

---

## 4.4 Discussion on Runtime

The computational complexity introduced by meta-learning is negligible in the context of few-shot expensive optimization. The time cost of model training mainly depends on the size of the dataset used for meta-learning (see Appendix J for a complexity analysis of meta-learning). In few-shot

optimization, the dataset is typically small, indicating that training meta-learning models is not time-consuming. In contrast, in expensive optimization, the time cost of each solution evaluation is much higher than the computational cost of model training. For example, each evaluation of engine performance in real-world engine calibration applications may take hours to days [55], similar to the evaluation costs reported in other studies on expensive optimization [45]. Therefore, in real-world applications, introducing a meta-learning model is worthwhile if even a single expensive evaluation can be saved.

## 5  Computational Studies

Our computational studies can be divided into three parts:

1. Appendix D evaluates our meta-learning model performance on two problems and analyzes model component contributions via ablation comparisons with model variants.
2. Sections 5.1 to 5.2 investigate the performance of our FSEO framework in enhancing sampling efficiency. Extensive ablation studies are conducted to provide guidance for practical applications of our FSEO framework.
3. Section 5.3 and Appendix H demonstrate the performance and broad applicability of our FSEO framework on real-world problems.

For all meta-learning methods used in our experiments, their basic setups are listed in Table 5. [1].

### 5.1  Performance on EMOPs

The experiment in this subsection is designed to answer the question below: With the experience learned from related tasks, can our FSEO framework help an SAEA save $9d$ solutions without a loss of optimization performance?

#### 5.1.1  Experimental Setups

**Optimizaion problems.** The computational study is conducted on DTLZ test problems [11]. All DTLZ problems have $d$=10 decision variables and 3 objectives, as the setups that have been widely used in [35]. The details of generating DTLZ variants (related tasks) are provided in Appendix C.

**Comparison algorithms.** We test our FSEO framework using an instantiation on MOEA/D-EGO, resulting in MOEA/D-FS. Details of the comparison algorithms are given in Appendix E.2.

**Optimization setups.** The parameter setups for this multi-objective optimization experiment are listed in Table 6. During the optimization process, an initial dataset $\mathcal{S}_*$ is sampled using Latin-Hypercube Sampling (LHS) method [28], then extra evaluations are conducted until the evaluation budget has run out. Note that we aim to use related tasks to save $9d$ evaluations without a loss of SAEA optimization performance. Hence, the total evaluation budgets for MOEA/D-FS and the comparison algorithms are different.

**Performance indicators.** Since the test problems have 3 objectives, we employ inverted generational distance plus (IGD+) [19] as our performance indicator, where smaller IGD+ values indicate better optimization results. 5000 reference points are generated for computing IGD+ values. More results in the metrics of IGD [5] and HV [61] are reported in Appendix E.4.

#### 5.1.2  Results and Analysis

The statistical test results are reported in Fig. 3 and Appendix E.3 (Table 7). It can be seen from Fig. 3 that, although 90 fewer evaluations are used in surrogate initialization, MOEA/D-FS can still achieve competitive or even smaller IGD+ values than MOEA/D-EGO on all DTLZ problems except for DTLZ7. In addition, the IGD+ values obtained by MOEA/D-FS drop rapidly, especially during the first few evaluations, implying the experience learned from DTLZ variants is effective. Therefore, in most situations, our FSEO framework is able to assist MOEA/D-EGO in reaching competitive or even better optimization results, with the number of evaluations used for surrogate initialization reduced from $10d$ to only $1d$.

---

[1]Code is available at https://github.com/XunzhaoYu/FSEO

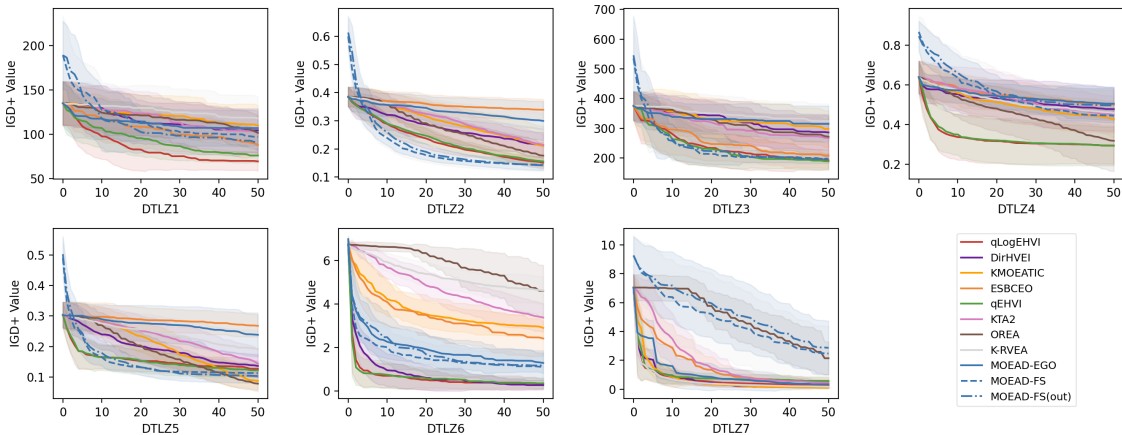

Figure 3: IGD+ curves averaged over 30 runs on 7 DTLZ problems. Solid lines are mean values, while shadows are error regions. MOEA/D-FSs and comparison algorithms initialize their surrogates with 10, 100 samples, respectively. X-axis denotes the extra 50 evaluations allowed in the further optimization. Note that 'FS(out)' indicates the target task is excluded from the range of related tasks during the meta-learning procedure (see Section 5.2.1).

MOEA/D-FS is less effective on DTLZ7 than on other DTLZ problems, which might be attributed to the discontinuity of the Pareto front on DTLZ7. Note that MOEA/D-FS learns experience from small datasets such as $\mathcal{D}_m$ and $\mathcal{S}_*$. The solutions in these small datasets are sampled at random, hence, the probability of having optimal solutions being sampled is small. However, it is difficult to learn the discontinuity of the Pareto front from the sampled non-optimal solutions. As a result, the knowledge of 'there are four discrete optimal regions' cannot be learned from such small datasets ($|\mathcal{D}_m| = 20$) collected from related tasks. The performance analysis between MOEA/D-FS and other comparison algorithms is available in Appendix E.3.

### 5.1.3 Further Comparison Experiments with Different Evaluation Budgets

We also compared the performance of our FSEO framework when only 10 evaluations are used for surrogate initialization for comparison algorithms. Consistent results are observed and reported in Table 10 in Appendix E.5. In addition, the performance of our FSEO framework in the context of extremely expensive optimization has been investigated in Appendix F (Table 11 and Fig. 7).

The question raised at the beginning of this subsection can be answered by the results discussed so far. Due to the integration of the experience learned from related tasks (DTLZ variants), although the evaluation cost of surrogate initialization has been reduced from $10d$ to $1d$, our FSEO framework is still capable of assisting regression-based SAEAs to achieve competitive or even better optimization results in most situations.

## 5.2 Ablation Studies on Influence of Task Similarity and Dataset Size in Meta-Learning

We conduct two ablation studies to investigate the influence of task similarity and that of the dataset size used in meta-learning.

### 5.2.1 Ablation Study: Influence of Task Similarity

In real-world applications, it is optimistic to assume that some related tasks are very similar to the target task. A more common situation is that all related tasks have limited similarity to the target task. To investigate the relationship between task similarity and FSEO optimization performance, we also test the performance in an 'out-of-range' situation, where the original DTLZ is excluded from the range of DTLZ variants during the MDKL meta-learning procedure. As a result, only the DTLZ variants that are quite different from the original DTLZ problem can be used to learn experience. The 'out-of-range' situation eliminates the probability that MDKL surrogates benefit greatly from the DTLZ variants that are very similar to the original DTLZ problem. Detailed definitions of the

related tasks used in the 'out-of-range' situation are given in Appendix C. Apart from the related tasks used, the remaining experimental setups are the same as the setups described in Section 5.1. For convenience, we denote the situation we tested in Section 5.1 as 'in-range' below.

Table 1: Mean IGD+ values and standard deviation (in parentheses) obtained from 30 runs on 7 DTLZ problems. Both MOEA/D-FSs initialize their surrogates with 10 samples, extra 50 evaluations are allowed in the further optimization. '+', '≈', and '−' denote the result of the 'out-of-range' situation is statistically significantly superior to, almost equivalent to, and inferior to that of the 'in-range' situation in the Wilcoxon rank sum test (significance level is 0.05), respectively. The last two rows count the statistical test results between MOEA/D-FSs and other compared algorithms.

| MOEA/D-FSs | In-range | Out-of-range |
|---|---|---|
| DTLZ1 | 9.70e+1(1.87e+1)≈ | 9.11e+1(1.53e+1) |
| DTLZ2 | 1.43e-1(2.29e-2)≈ | 1.41e-1(1.75e-2) |
| DTLZ3 | 1.97e+2 (1.64e+1)≈ | 1.98e+1(1.51e+1) |
| DTLZ4 | 4.44e-1(1.35e-1)≈ | 4.96e-1(8.63e-2) |
| DTLZ5 | 1.13e-1(2.24e-2)≈ | 1.03e-1(2.39e-2) |
| DTLZ6 | 1.11e+0(5.71e-1)≈ | 1.17e+0(6.88e-1) |
| DTLZ7 | 2.47e+0(1.89e+0)≈ | 2.86e+0(1.87e+0) |
| $+/\approx/-$ | 0/7/0 | -/-/- |
| vs MOEA/D-EGO | 4/2/1 | 4/2/1 |
| vs 9 Comparisons | 30/15/18 | 31/13/19 |

The statistical test results reported in Table 1 show that the 'out-of-range' situation achieves competitive IGD+ values to the 'in-range' situation on all 7 test instances. This suggests that related tasks that are very similar to the target task have a limited impact on the optimization performance of our FSEO framework. Useful experience can be learned from related tasks that are not very similar to the target task. Crucially, when comparing the performance of the 'out-of-range' situation and that of MOEA/D-EGO, we can still observe competitive or improved optimization results on 6 DTLZ problems (see Table 1, the row titled by 'vs MOEA/D-EGO', or Fig. 3). Moreover, it can be seen from the last row of Table 1 that the 'out-of-range' situation achieves better/competitive/worse IGD+ values than all compared SAEAs on 31/13/19 test instances. In comparison, the corresponding statistical test results for the 'in-range' situation are 30/15/18. The difference between these statistical test results is not significant.

A study on the 'out-of-range' situation in the context of extremely expensive multi-objective optimization is presented in Appendix F.2. Consistent results are observed in Table 12 and Fig. 7.

Consequently, related tasks that are very similar to the target task are not essential to the optimization performance of our FSEO framework. In the 'out-of-range' situation, our MOEA/D-FS can still achieve competitive or better optimization results than MOEA/D-EGO while using only $1d$ samples for surrogate initialization.

### 5.2.2 Ablation Study: Influence of the Size of Datasets Used in Meta-Learning

We also investigated the performance of our FSEO framework when different sizes of datasets $|\mathcal{D}_m|$ are used in the meta-learning procedure. The experimental setups are the same as the setups of MOEA/D-FS in Section 5.1 except for $|\mathcal{D}_m|$.

It is evident from Table 2 that when each DTLZ variant provides $|\mathcal{D}_m| = 60$ samples for the meta-learning of MDKL surrogates, the performance of both MOEA/D-FSs are improved on 2 or 3 DTLZ problems. Particularly, a significant improvement can be observed from the optimization results of DTLZ7. As we discussed in Section 5.1, the poor performance of our experience-based optimization on DTLZ7 is caused by the small size of $\mathcal{D}_m$. Optimal solutions have few chances to be included in a small $\mathcal{D}_m$, which makes $\mathcal{D}_m$ fail to provide the experience about the discontinuity of optimal regions. In comparison, the experience of 'optimal regions' can be learned from large datasets $\mathcal{D}_m$ and thus the optimization results are improved significantly.

In conclusion, for our FSEO framework, a large $\mathcal{D}_m$ for the meta-learning procedure indicates more useful experience can be learned from related tasks, which further improves the performance of experience-based optimization. Therefore, when applying our FSEO framework to real-world optimization problems, it is preferable to collect more data from related tasks for experience learning.

### 5.3 Performance on Real-World Problems

We also evaluate the performance of our FSEO on real-world problems. In this section, we focus on a Network Architecture Search (NAS) problem, and more computational studies on real-world problems are reported in Appendix H. This NAS problem optimizes the architecture of a Transformer

Table 2: Mean IGD+ values and standard deviation (in parentheses) obtained from 30 runs on 7 DTLZ problems. 10 samples are used for initialization and extra 50 evaluations are allowed in the further optimization. $|\mathcal{D}_m|$ is the size of the dataset collected from each related task.

| Problem | In-range | | Out-of-range | |
| --- | --- | --- | --- | --- |
| | $|\mathcal{D}_m|$=20 | $|\mathcal{D}_m|$=60 | $|\mathcal{D}_m|$=20 | $|\mathcal{D}_m|$=60 |
| DTLZ1 | 9.70e+1(1.87e+1)≈ | 9.77e+1(1.73e+1) | 9.11e+1(1.53e+1)≈ | 9.93e+1(1.87e+1) |
| DTLZ2 | 1.43e-1(2.29e-2)+ | 1.24e-1(2.11e-2) | 1.41e-1(1.75e-2)+ | 1.29e-1(2.36e-2) |
| DTLZ3 | 1.97e+2 (1.64e+1)≈ | 1.98e+2 (2.21e+1) | 1.98e+1(1.51e+1)≈ | 1.93e+2(1.19e+1) |
| DTLZ4 | 4.44e-1(1.35e-1)≈ | 5.17e-1(5.68e-2) | 4.96e-1(8.63e-2)≈ | 5.17e-1(5.38e-2) |
| DTLZ5 | 1.13e-1(2.24e-2)+ | 9.96e-2(2.18e-2) | 1.03e-1(2.39e-2)≈ | 1.05e-1(2.73e-2) |
| DTLZ6 | 1.11e+0(5.71e-1)≈ | 1.04e+0(6.06e-1) | 1.17e+0(6.88e-1)≈ | 1.22e+0(6.41e-1) |
| DTLZ7 | 2.47e+0(1.89e+0)+ | 7.49e-1(2.61e-1) | 2.86e+0(1.87e+0)+ | 6.96e-1(2.41e-1) |
| $+/\approx/-$ | 3/4/0 | -/-/- | 2/5/0 | -/-/- |

in terms of two objectives: error and flops. Fig. 4 illustrates the result, detailed experimental setups are available in Appendix G.

Fig. 4 illustrates the optimization results in terms of Hypervolume (HV) values, a large HV value indicates a good performance. We can observe that MOEA/D-FS, qLogEHVI, K-RVEA, and DirHVEI are preferable to the remaining comparison algorithms in this NAS problem. However, we should note that MOEA/D-FS uses only $1d$ samples from the target task to initialize surrogates. Due to the small initialization dataset, at the early stage, the initial HV value of MOEA/D-FS is smaller than the initial HV values of other comparison algorithms (see Fig. 4). With our meta-learning models, MOEA/D-FS adapts to the target task rapidly and it achieves a competitive

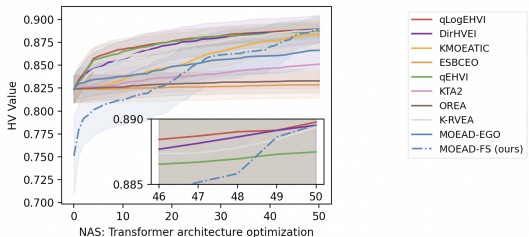

Figure 4: NAS comparison results. MOEA/D-FS and comparison algorithms initialize their surrogates with 18, 100 samples, respectively. MOEA/D-FS reaches competitive results while 82 evaluations are saved.

HV value within 50 additional evaluations, which is a substantial improvement in optimization performance when compared to the performance of its underlying example algorithm, MOEA/D-EGO. This implies that MOEA/D-FS has saved 82 more evaluations than comparison algorithms by learning experience from related tasks and also has improved the performance of underlying optimization algorithm simultaneously. Therefore, the effectiveness of our FSEO framework on this real-world EMOP is demonstrated.

## 6   Conclusion and Future Work

**Conclusion.** We present a FSEO framework to address EMOPs from the perspective of SAEAs. A novel meta-learning approach MDKL is proposed to learn prior experience from related expensive tasks. MDKL model is designed for optimization and has explicit task-specific parameters, which allows continually update of task-specific parameters during the optimization process. Empirical experiments show that the FSEO framework is able to improve sampling efficiency and save expensive evaluations for existing regression-based SAEAs. Ablation studies reveal the influence between optimization performance and solutions similarity as well as the size of datasets for meta-learning.

**Limitation and future work.** The limitations of this work can be summarized as the following two points, they widely exist in the literature: First, we do not have a mathematical definition of related tasks. As a result, the boundary between related and unrelated tasks is not clear, making it difficult to perform theoretical analysis on task similarity. Second, the proposed framework is currently for regression-based SAEAs only. A detailed discussion on this point is available in Appendix B.

Future work could focus on quantifying task similarity by proposing a metric to measure similarity between tasks. With an appropriate task similarity measure, systematic studies on few-shot optimization and experience-based optimization could be conducted. In addition, a few-shot optimization framework for other SAEA categories can also be a future work.

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
