# OpenReview forum: "FSEO: Few-Shot Evolutionary Optimization via Meta-Learning for Expensive Multi-Objective Optimization"
_NeurIPS.cc/2025/Conference — NeurIPS 2025 poster_

### Official Review · Reviewer_jgNV · 2025-06-29

**Clarity:** 3
**Significance:** 4
**Originality:** 4
**Rating:** 6
**Confidence:** 3

**Summary:**

This paper introduces FSEO, a framework for solving expensive multi-objective optimization problems (EMOPs) using few-shot meta-learning. At its core, the method builds a meta-learned surrogate model (MDKL) that combines a deep kernel architecture with Gaussian Processes. The surrogate is trained across a set of related tasks to learn task-independent parameters and then adapted online to each new target task with limited samples (as few as 1d evaluations). The surrogate is integrated into regression-based surrogate-assisted evolutionary algorithms (SAEAs), with MOEA/D-EGO used as a case study.
The authors validate their approach across synthetic DTLZ benchmarks and real-world tasks, including neural architecture search and engine calibration, and conduct ablation studies to explore the effects of task similarity and dataset size.

**Questions:**

1. Without an acquisition function, how does the evolutionary component balance uncertainty? Is the MDKL variance ever used directly in search?
2. Do you have any quantitative measures of task similarity across DTLZ variants or real tasks (e.g., using function landscapes or mutual information)?
3. How expensive is the adaptation step computationally? Could you report runtime per update or total time vs. baseline?
4. In DTLZ7, you mention MDKL fails due to discontinuity. Have you considered hybrid approaches (e.g., combining meta-learned GP with density estimators)?
5. Have you tried plugging MDKL into other SAEAs like RVEA or ParEGO? If not, are there any anticipated challenges?


Suggestions for Improvement

* Add a second SAEA to test generality of the FSEO framework.
* Provide runtime measurements (total time, per-update cost, adaptation vs. inference).
* Discuss whether task-specific adaptation ever overfits in low-data settings.
* Add theoretical discussion or intuition about why MDKL generalizes across tasks.

**Ethical Concerns:**

["NO or VERY MINOR ethics concerns only"]

**Final Justification:**

My concerns have all been resolved, so I’ll keep my score as it is.

**Limitations:**

yes

**Quality:**

4

**Strengths And Weaknesses:**

Strengths

* The need for few-shot sample-efficient optimization is strong, especially in high-cost engineering and machine learning settings.
* FSEO is a modular enhancement that can be plugged into any regression-based SAEA, not tied to a specific optimizer.
* MDKL introduces task-specific parameter adaptation during optimization, improving over frozen meta-learned surrogates used in past few-shot BO literature.
* Comprehensive evaluation and the "out-of-range" setting shows that FSEO remains effective even when test tasks fall outside the meta-training distribution.

Weaknesses

* Performance inconsistency: Results on DTLZ4, DTLZ6, and especially DTLZ7 show that FSEO can struggle with non-uniform or discontinuous Pareto fronts — an important limitation.
* No theoretical guarantees: The paper does not provide formal analysis (e.g., regret bounds, convergence guarantees) to support when/why MDKL works or fails.
* Generalization to other SAEAs not demonstrated: All experiments focus on MOEA/D-EGO. Claims of framework generality would be stronger if results on a second SAEA were included.
* Compute/resource use is under-reported: Adaptation cost per iteration and overall runtime are not measured, even though MDKL retraining happens repeatedly.

---

> ### Author Rebuttal · Authors · 2025-07-31
>
> Question 1
>
> Good point.
> Our framework includes meta-learning models and a model update strategy. However, the acquisition function is a component of the optimizer and can vary depending on which optimization algorithm the FSEO framework is applied to.
>
> - - -
> Question 2
>
> We estimate task similarity across variants based on the difference between their parameters, following the approach used in several classic meta-learning studies [1-2]. As suggested by the reviewer, using function landscapes could be a promising idea in this regard. We will add figures to analyze the task similarity across variants based on the difference of function landscapes.
>
> >[1] Finn, Chelsea, et al. ”Model-Agnostic Meta-Learning for Fast Adaptation of Deep Networks." ICML, 2017
>
> >[2] Patacchiola, Massimiliano, et al. “Bayesian Meta-Learning for the Few-Shot Setting via Deep Kernels." NeurIPS, 2020.
>
> - - -
> Question 3
>
> We will revise our paper to include the following discussion on runtime.
>
> The computational cost of the adaptation step is minimal, and we did not observe a noticeable increase in runtime during the optimization process. This is expected because:
> 1. We did not increase the number of surrogate models in SAEAs.
> 2. No additional data from the target task was introduced during the adaptation step.
> 3. In the context of few-shot optimization, the dataset used for adaptation is small, so the model updates are not computationally intensive.
>
> Overall, in expensive optimization problems, the cost of evaluating a solution typically outweighs the runtime of the optimization algorithm itself.
>
> - - -
> Question 4
>
> This is a great suggestion.
> As a pioneering study on multi-objective few-shot optimization, our current work does not yet explore additional techniques to address the limitations of meta-learning models on problems with specific characteristics (e.g., discontinuities). We will revise our paper to clarify that enhancing optimization performance through methods such as hybrid approaches is a promising direction for future work.
>
> - - -
> Question 5:
>
> Good suggestion.
> We should make it clearer that we integrated MDKL with another SAEA in Section 5.3.2. This section is intended to demonstrate the generality and broad applicability of our algorithm, and we did not encounter any unexpected challenges.
>
> - - -
> Weakness 1:
>
> We should make it clearer that although FSEO did not achieve better optimization results on DTLZ4 and DTLZ6, the results were obtained using only 1\$d$ evaluations, indicating 9\$d$ evaluations were saved. This reflects a significant improvement in efficiency for expensive optimization.
> Additionally, the results in Table 9 (Appendix E.5) further demonstrate that FSEO performs well on DTLZ4 and DTLZ6.
>
> - - -
> Weakness 2 (Suggestion 4):
>
> Good suggestion. We will revise our paper and add the following intuition to explain why MDKL generalizes across tasks.
>
> For related tasks, we assume that the parameters of their distributions are located within a cluster in the parameter space. If two parameter vectors are very close to each other in this space, the corresponding tasks are considered highly similar. Therefore, the purpose of the meta-learning procedure is to move the task-independent parameter vector toward the center of this cluster. Once meta-learning is complete, it becomes easier to approximate the parameter vector of the target distribution based on the position of the meta-learned parameter vector.
>
> - - -
> Weakness 3 (Suggestion 1):
>
> See our response to Question 5.
>
> - - -
> Weakness 4 (Suggestion 2):
>
> See our response to Question 3.

---

### Official Review · Reviewer_jiRL · 2025-07-01

**Clarity:** 3
**Significance:** 3
**Originality:** 3
**Rating:** 4
**Confidence:** 3

**Summary:**

This paper introduces FSEO, a few-shot optimization framework based on meta-learning to solve expensive multi-objective optimization problems (EMOPs). The proposed framework uses a novel meta-learning surrogate model called Meta Deep Kernel Learning (MDKL), which combines neural networks with Gaussian Processes. The framework aims to improve the sample efficiency of surrogate-assisted evolutionary algorithms (SAEAs) by leveraging prior experience learned from small datasets of related tasks
A meta learned surrogate that enables continual adaptation to new tasks
A generalized framework for use in few-shot scenarios
framework is shown to reduce the number of function evaluations needed while achieving comparable or better optimization performance compared to standard methods

**Questions:**

Please refer to the strengths and weakness section to address questions.

**Ethical Concerns:**

["NO or VERY MINOR ethics concerns only"]

**Final Justification:**

I went through the reviewers comments, and I am upgrading my review.

**Limitations:**

Please refer to the strengths and weakness section to address questions.

**Quality:**

3

**Strengths And Weaknesses:**

Strengths
The paper clearly targets a few-shot setting for expensive multi-objective optimization which is still an active area of research.  MDKL supports continual adaptation of task-specific parameters, which is architecturally significant.  Experiments were conducted on real world tasks where significant improvements have been shown. he paper includes controlled ablations for Task similarity (in-range vs. out-of-range).

Weakness:

There is missing comparisons for other recent meta learning baselines.  There is no comparison to other meta-learning frameworks that target optimization (e.g., meta-acquisition functions or meta-BO on EMOPs), such as MALIBO, PREBO, or transfer-SAEA methods
Synthetic results are only reported on DTLZ problems (DTLZ1–DTLZ7), which are standard but limited, No large-scale or high-dimensional EMOP benchmarks.
There are no much information on hyperparameters for reproduciability purpose, please direct me in rebuttal if I am missing them
While IGD+ curves and error bands are provided in synthetic benchmarks (Figure 3), no error bars or statistical significance tests are shown.

---

> ### Author Rebuttal · Authors · 2025-07-31
>
> Weakness 1:
>
> Thank you for pointing out this issue.
> We should have made it clearer that we actually have comparisons with meta-learning baselines in our modeling experiments (see Appendix D).
> It is worth noting that many few-shot optimization methods use similar meta-learning models but differ in the strategies built around them. Additionally, some studies use the term 'meta-learning' more broadly to refer to all experience-based optimization methods that leverage data from related tasks (e.g., [1]). As a result, these 'meta-learning' methods may not serve as direct baselines in the specific context of few-shot optimization.
> A discussion on the differences between various experience-based optimization methods is provided in Appendix A.1.
>
> > [1] Wang, Zi, et al. ”Pre-Trained Gaussian Processes for Bayesian Optimization." Journal of Machine Learning Research, 2024.
>
> - - -
> Weakness 2:
>
> This is a good suggestion. We have read the recommended papers and conducted additional research on this topic.
>
> We should clarify that our work focuses specifically on few-shot multi-objective optimization. While some existing studies on meta-learning-based optimization exist (e.g., MALIBO), they primarily target single-objective optimization or other types of experience-based optimization. As such, these frameworks are not directly comparable to our approach in the context of multi-objective optimization problems.
> We have included a discussion on the distinctions among various experience-based optimization methods in Appendix A.1, such as multi-task optimization, multi-form optimization, transfer optimization, and few-shot optimization.
>
> Related work on multi-objective optimization is very limited, and some of them are limited on specific optimization applications [1], which motivated us to develop a generalized framework for multi-objective optimization.
>
> > [1] Watanabe, Shuhei, et al. "Speeding up multi-objective hyperparameter optimization by task similarity-based meta-learning for the tree-structured parzen estimator." IJCAI, 2023.
>
> - - -
> Weakness 3:
>
> Great point.
> The DTLZ benchmark includes seven problems with different but common features, which helps evaluate the performance of multi-objective optimization algorithms. However, as the reviewer noted, these benchmark problems may not fully capture the complexity of real-world applications.
>
> To address this limitation and strengthen our empirical studies, we followed the practice in the literature by including several real-world problems to demonstrate the robustness of our method. The results on these real-world applications are reported in Section 5.3.
> Additionally, our modeling experiments in Appendix D include both synthetic and real-world applications.
>
> - - -
> Weakness 4:
>
> Thank you for pointing out this issue.
> As a pioneering study in few-shot multi-objective optimization, our work focuses on designing an efficient and general framework for the most common multi-objective optimization scenarios found in real-world expensive optimization applications. Accordingly, we followed the practice of existing studies [1-4] and did not include specific problem features such as large-scale or dynamic settings within the scope of this work.
>
> We understand the importance of considering higher-dimensional problems, and we will leave it for one subsequent work (just like other studies that focus on this challenging issue [5-7]). We will make this clear in the revised paper.
> >[1] Wang, Zi, et al. ”Pre-Trained Gaussian Processes for Bayesian Optimization." Journal of Machine Learning Research, 2024.
>
> >[2] Pan, Jiarong, et al. “MALIBO: Meta-Learning for Likelihsood-Free Bayesian Optimization." ICML, 2024.
>
> >[3] Tu et al. ”Joint Entropy Search for Multi-Objective Bayesian Optimization.” NeurIPS, 2022.
>
> >[4] Wistuba, Martin, and Josif Grabocka. “Few-Shot Bayesian Optimization with Deep Kernel Surrogates." ICLR, 2021.
>
> > [5] Wu, Yupeng, et al. “High-Dimensional Causal Bayesian Optimization.” ECAI, 2024.
>
> > [6] Ramchandran, Siddharth, et al. “High-Dimensional Bayesian Optimisation with Gaussian Process Prior Variational Autoencoders.” ICLR, 2025.
>
> > [7] Zhou, MengChu, et al. “Evolutionary Optimization Methods for High-Dimensional Expensive Problems: A Survey.” IEEE/CAA Journal of Automatica Sinica, 2024.
>
> - - -
> Weakness 5:
>
> Thank you for your comment.
> We provide the hyperparameter details in Tables 4 and 5 (Appendix E.1), and we have cited relevant references to show that these settings are commonly used in the literature.
> We will revise the paper to clarify this point more effectively.
>
> - - -
> Weakness 6:
>
> Thanks for your comment.
> Statistical tests corresponding to Figure 3 are reported in Appendix E.3, Table 6.
> In addition, statistical test results for two different metrics, IGD and HV, are presented in Tables 7 and 8.
> We will revise our paper to improve clarity and make key information easier to find.

---

### Official Review · Reviewer_Z9sy · 2025-07-03

**Clarity:** 2
**Significance:** 2
**Originality:** 3
**Rating:** 4
**Confidence:** 4

**Summary:**

The paper proposes a few-shot optimization approach for multi-objective optimization. It consists in learning a parameterized deep kernel learning model that learn surrogate for optimization. The approach is using the meta-learning procedure to estimate the parameters of the deep kernel learning to from the few samples available so far, in order to achieve good performance for multi-objective optimization. The approach is evaluated first on synthetic multi-objective optimization benchmark (DTLZ) and then on two more realistic problems (neural architecture search and engine calibration). Results show performance on par with the best approaches for each problem, while reducing the number of samples used for reaching good solutions for the optimization tasks.

**Questions:**

- Can you provide a clear context, providing a better overview of related work and how the current contribution differ from related work and constitue a meaningful contribution to the field? I found the discussion in Sec. 2 to be difficult to follow and not that helpful to proper position the work, there is a lot of very specific details without provide a clear high-level picture.
- Can you provide a clear statement of the significance of the proposal and the reach of it? Said otherwise, “why should I care” about the current proposal.

**Ethical Concerns:**

["NO or VERY MINOR ethics concerns only"]

**Final Justification:**

I have read the rebuttal comments from the authors. These are fine but are not providing significant points that would change my judgement on the paper. I maintain my evaluation.

**Limitations:**

The proposal is quite algorithmic and has not clear and direct limitations and negative impact on society. Therefore, that point is not applicable to the paper.

**Paper Formatting Concerns:**

Mostly fine.

**Quality:**

3

**Strengths And Weaknesses:**

Strengths:
- A sound proposal for few shot optimization, the overall method makes sense.
- Good set of experiments to characterize the approach.
- Results are generally comparable to state-of-the-art, except for some cases.

Weaknesses:
- Rather niche topic, the impact of the proposal is rather limited.
- The experimental methodology is dubious, as the variants used to train the model are based on DTLZ benchmark, which is then used to assess the model. There are leakage risks, and performances are probably overestimated. I think the meta-training / meta-testing should be done over distincts DTLZ problems, the one used in training not be used again in testing. A leave-one-out methodology over can allow to make a better evaluation. Even with the extra experiments with task similarity used to avoid using close tasks for training-testing, I still think there are potential leakage issues that reduce confidence we can have over the results.
- The proposal is not improving performances much, so it is not clear that the extra complexity added with the meta-learning worth it.
- Related work section is not very helpful, not providing clear context to position the work.
- Overall text quality can be improved, the writing is not always very clear. Careful proofreading would improve the quality of the paper.

---

> ### Author Rebuttal · Authors · 2025-07-31
>
> Weakness 1:
>
> The topic of addressing expensive optimization problems using meta-learning has gained popularity in recent years [1-2]. However, most existing studies focus on the simplest setting: Single-objective optimization, while other common scenarios such as multi-objective optimization remain underexplored.
> Furthermore, many of these works are specific and lack generalization capability.
> In contrast, our work introduces an efficient meta-learning model and a generalized optimization framework for expensive multi-objective optimization. In our humble opinion, it has the potential to make a broader impact than previous related studies.
>
> >[1] Wang, Zi, et al. ”Pre-Trained Gaussian Processes for Bayesian Optimization." Journal of Machine Learning Research, 2024.
>
> >[2] Pan, Jiarong, et al. “MALIBO: Meta-Learning for Likelihood-Free Bayesian Optimization." ICML, 2024.
>
> - - -
> Weakness 2:
>
> Our experimental methodology follows the methodology used in several classic meta-learning studies [1-2].
> Essentially, meta-learning aim to extract common features from related tasks. However, different DTLZ problems often have distinct features. For example, DTLZ3 is a multimodal problem while DLTZ4 is unimodal but scaled. Therefore, it may not be appropriate to treat DTLZ3 and DTLZ4 as related tasks.
> In our humble opinion, the methodology suggested by the reviewer might be more aligned with transfer learning rather than meta-learning. A discussion on the difference between transfer learning and meta-learning is available in Appendix A.1.
> Note that some studies [3] use the term 'meta-learning' more broadly to refer to all experience-based optimization methods that leverage data from related tasks. However, in this work, we adopt the definition of meta-learning as presented in [1].
>
> >[1] Finn, Chelsea, et al. "Model-Agnostic Meta-Learning for Fast Adaptation of Deep Networks." ICML, 2017.
>
> >[2] Patacchiola, Massimiliano, et al. "Bayesian Meta-Learning for the Few-Shot Setting via Deep Kernels." NeurIPS, 2020.
>
> >[3] Wang, Zi, et al. ”Pre-Trained Gaussian Processes for Bayesian Optimization." Journal of Machine Learning Research, 2024.
> - - -
>
> Weakness 3:
>
> Good point. We have revised our manuscript and added the following discussion to further clarify the relationship between performance and the complexity of meta-learning:
>
> The computational complexity introduced by meta-learning is negligible in the context of expensive optimization. Since the dataset used for meta-learning is typically small, training the meta-learning model is not time-consuming. In contrast, evaluating a single solution can be extremely costly. For example, an engine performance evaluation in our real-world application may take hours to days, similar to the evaluation costs reported in other studies on expensive optimization [1].
> Therefore, in real-world applications, introducing a meta-learning model is worthwhile if even a single expensive evaluation can be saved.
>
> From our experiments, it is evident that our few-shot optimization framework saves 9\$d$ evaluations, while still achieving competitive, or even superior, optimization results (e.g., in terms of IGD+ values) in most cases.
>
> >[1] Wang, Zi, et al. “Pre-Trained Gaussian Processes for Bayesian Optimization." Journal of Machine Learning Research, 2024.
>
> - - -
>
> Weakness 4:
>
> See our response to Question 1.
>
> - - -
>
> Weakness 5:
>
> Thanks for your suggestion. We have carefully gone through the entire manuscript to improve the text quality. We believe the revised version is clearer than the one initially submitted.
>
> - - -
>
> Question 1
>
> Thank you for your suggestion. We have revised the related work section and added the following overview to the second paragraph of Section 2.
>
> Wistuba firstly employed meta-learning for few-shot optimization on hyperparameter optimization (HPO) problems. Subsequent related studies can be grouped into three categories:
>
> 1. The first category focuses on improving the performance of few-shot optimization, either by employing different models [2] or by developing novel acquisition functions for BO [3].
>
> 2. The second category extends few-shot optimization to more complex optimization problems [4] or applies it to new domains [5-6].
>
> 3. The third category explores the use of related tasks in other experience-based optimization methods, such as transfer optimization [7] and multi-task optimization [8], which are distinct from FSO.
>
> However, many studies use existing meta-learning models from the literature, these models are not designed for optimization purpose. As a result, these models may be less efficient when applied to optimization problems. In addition, many acquisition functions are working for specific BO, limiting their generality.
> Moreover, current research on few-shot optimization focuses primarily on the simplest case, single-objective optimization. However, multi-objective optimization is common in real-world applications and presents greater challenges.
>
> In response to these gaps, our work proposes an efficient meta-learning model for optimization purpose and a generalized framework with broad applicability. In particular, we aim to fill the gap in multi-objective few-shot optimization.
> The difference between our work and related work is described in the third paragraph of Section 2.
>
> >[1] Wistuba, Martin, and Josif Grabocka. “Few-Shot Bayesian Optimization with Deep Kernel Surrogates." ICLR, 2021.
>
> >[2] Maraval, Alexandre, et al. "End-to-End Meta-Bayesian Optimization with Transformer Neural Processes." NeurIPS 2023.
>
> >[3] Hsieh, Bing-Jing, et al. ”Reinforced Few-Shot Acquisition Function Learning for Bayesian Optimization." NeurIPS, 2021.
>
> >[4] Zhang, Huan, et al. “Solving Expensive Optimization Problems in Dynamic Environments with Meta-Learning." IEEE Trans on Cybernetics, 2024.
>
> >[5] Chakrabarty, Ankush. ”Optimizing Closed-Loop Performance with Data from Similar Systems: A Bayesian Meta-Learning Approach." CDC, 2022.
>
> >[6] Chen, Wenlin, et al. “Meta-learning Adaptive Deep Kernel Gaussian Processes for Molecular Property Prediction." NeurIPS, 2022.
>
> >[7] Bai, Tianyi, et al. ”Transfer Learning for Bayesian Optimization: A Survey." arXiv, 2023.
>
> >[8] Wang, Zi, et al. “Pre-Trained Gaussian Processes for Bayesian Optimization." Journal of Machine Learning Research, 2024.
>
> - - -
> Question 2
>
> In short, the significance of our work can be summarized as follows:
>
> 1. We fill the gap in multi-objective optimization for few-shot optimization, which is crucial as multi-objective optimization problems widely exist in real-world applications.
>
> 2. We propose a novel meta-learning model specifically designed for optimization. It can be efficiently and continuously adapted during the optimization process.
>
> 3. Our FSEO framework is compatible with a variety of optimization algorithms and is not limited to a specific Bayesian optimization method.

---

> > ### Comment · Reviewer_Z9sy · 2025-08-07
> >
> > I have read the rebuttal comments from the authors. These are fine but are not providing significant points that would change my judgement on the paper. I maintain my evaluation.

---

### Official Review · Reviewer_Nr53 · 2025-07-03

**Clarity:** 2
**Significance:** 2
**Originality:** 2
**Rating:** 4
**Confidence:** 3

**Summary:**

In this paper, the authors propose a meta-learning-based approach for few-shot, expensive multi-objective optimization problems. The method leverages the meta-learning framework, which is well-suited for few-shot scenarios, and introduces additional components to handle the challenges of multi-objective optimization. A kernel function is used to learn from related tasks and is integrated with a neural network to capture more complex relationships. The learned knowledge is then transferred to task-specific parameters. Additionally, the authors propose a surrogate update strategy based on the mean squared error (MSE) of a given surrogate model. If the updated surrogate performs worse than the previous one, the model reverts to the earlier version. To demonstrate the effectiveness of their approach, the authors present extensive numerical results.

**Questions:**

How many related tasks are needed to train a good model, and how long does it take?

Can you explain more on the model-parameter adapt and data adapt part?

**Ethical Concerns:**

["NO or VERY MINOR ethics concerns only"]

**Limitations:**

yes

**Paper Formatting Concerns:**

No issue.

**Quality:**

3

**Strengths And Weaknesses:**

Strengths:
1: The authors effectively adopt the meta-learning framework to address multi-objective optimization problems.
2: The surrogate update strategy is interesting, as it introduces a new way to evaluate update directions and discard less effective ones.
3: Compared with the vanilla meta learning, the author more effectively uses prior experience.

Weakness:
1:The meta-learning approach is computationally intensive. The authors should provide a discussion of the associated computational cost and training time.
2: The author claims this method is the model parameter adpation. However, the meta learning itself also a model adaption. I don't see the novelty claimed in this paper. It is more like an more effective meta learning method. It will be helpful if author can explain more on that.

---

> ### Author Rebuttal · Authors · 2025-07-31
>
> Weakness 1
>
> Good point. We will revise our paper to highlight the following discussion.
>
> In expensive optimization, the time cost of each solution evaluation is much higher than the computational cost of model training.
> At the end of Section 5.3.2, we provide a brief discussion on runtime using a real-world engine application as an example, illustrating how costly an engine performance evaluation can be.
> In contrast, the time cost of model training mainly depends on the size of the dataset used for meta-learning (see Appendix L for a complexity analysis of meta-learning). However, in the context of few-shot optimization, the dataset size is small, indicating that the computational cost of model training is trivial compared to the cost of solution evaluations.
>
> - - -
> Weakness 2
>
> Indeed, one of our novelties is a more effective meta-learning method which has a novel architecture for continual parameter adaptation. The key point is when and how to adapt models.
>
> Existing meta-learning based optimization methods use models that meta-learned for regression purpose rather than optimization. As a result, the model parameters are adapted only during the meta-learning process. In the optimization process, these models are updated with new data collected from the target task but no further adaptation of model parameters typically occurs.
> One of our novelties is the meta-learning model architecture that designed for optimization purpose, this architecture allows the model parameters to be continuously adapted throughout the optimization process.
>
> - - -
> Question 1:
>
> The performance of the model depends on several factors, including the number of related tasks, the similarity among those tasks, and the amount of data available for each task. If we define a ‘good' model as one that can effectively improve the optimization efficiency of a target task, then even a small number of related tasks (e.g., 5-10) may be sufficient. However, the actual improvement in efficiency also strongly depends on factors such as task similarity and the quantity of data within each task.
>
> As for the training time, it would not take a long time due to the small size of datasets in the context of few-shot optimization. In our empirical studies, the meta-learning process takes only a few minutes. Compared to the costly evaluation in expensive optimization (hours to days), the time cost of meta-learning is negligible.
>
> - - -
> Question 2:
>
> The parameters in our meta-learning model consist of two parts: task-independent parameters and task-specific parameters (as illustrated in Fig. 1 of our paper).
> During the meta-learning procedure, the task-independent parameters are meta-learned using related tasks.
> In the optimization procedure, the task-specific parameters are adapted using data from the target task.
>
> In contrast, meta-learning models originally designed for regression tasks do not include explicit task-specific parameters. In their optimization process, adaptation is typically achieved by updating the data used to compute the correlation matrix $R$ (see Equation 4 in our paper), rather than by adjusting model parameters.
>
> Please let us know if any part remains unclear or which specific point you are confused about, and we would be happy to clarify further.

---

> > ### Comment · Reviewer_Nr53 · 2025-08-03
> >
> > Thanks for author's detailed response. I would keep my current judge as borderline accept.

---

### Decision · Program_Chairs · 2025-09-17

**Decision:**

Accept (poster)

**Comment:**

This paper develops a novel meta-learning model for optimization purposes and proposes a generalized few-shot evolutionary optimization (FSEO) method for expensive multi-objective optimization problems. The proposed FSEO uses meta deep kernel learning (MDKL), which combines neural networks with Gaussian processes (GP). The surrogate is trained across related tasks to learn task-independent parameters and adapted to a new target task. The experimental evaluation on benchmark problems demonstrates that FSEO outperforms baseline methods.

The proposed meta-learning framework can adequately address the challenges in expensive multi-objective optimization problems. The weaknesses are that the proposed method requires a relatively high computational cost and lacks a theoretical discussion.

After the authors' rebuttal and discussion, all reviewers gave a positive score. The proposed meta-learning framework and surrogate update strategy are sound. The effectiveness of the proposed method is empirically validated through experiments.
The weaknesses and room for improvement might exist in this paper, which might be a reason that several reviewers did not raise the scores. However, based on the reviewers' comments and discussion, I believe that the strengths of this paper outweigh its weaknesses. Therefore, I recommend accepting this paper.

I encourage the authors to revise the paper based on the reviewers' comments to improve the quality of the paper, e.g., the clarity of the novelty of the proposed method, justification of the experimental settings, and the extra computational cost.